# ON SUMMARIZED VALIDATION CURVES AND GENERALIZATION

## ABSTRACT

The validation curve is widely used for model selection and hyper-parameter search with the curve usually summarized over all the training tasks. However, this summarization tends to lose the intricacies of the per-task curves and it isn't able to reflect if all the tasks are at their validation optimum even if the summarized curve might be. In this work, we explore this loss of information, how it affects the model at testing and how to detect it using interval plots. We propose two techniques as a proof-of-concept of the potential gain in the test performance when per-task validation curves are accounted for. Our experiments on three large datasets show up to a 2.5% increase (averaged over multiple trials) in the test accuracy rate when model selection uses the per-task validation maximums instead of the summarized validation maximum. This potential increase is not a result of any modification to the model but rather at what point of training the weights were selected from. This presents an exciting direction for new training and model selection techniques that rely on more than just averaged metrics.

## 1 INTRODUCTION

A validation set, separate from the test set, is the de facto standard for training deep learning models through early stopping. This non-convergent approach (Finnoff et al., 1993) identifies the best model in multi-task/label settings based on an expected error across all tasks. Calculating metrics on the validation set can estimate the model's generalization capability at every stage of training and monitoring the summarized validation curve over time aids the detection of overfitting. It is common to see the use of validation metrics as a way to stop training and/or load the best model for testing, as opposed to training a model to N epochs and then testing. While current works have always cautioned about the representativeness of validation data being used, the curves themselves haven't been addressed much. In particular, there hasn't been much attention on the summarized nature of the curves and their ability to represent the generalization of the constituent tasks.

Tasks can vary in difficulty and even have a dependence on each other (Graves, 2016; Alain & Bengio, 2016). An example by Lee et al. (2016) is to suppose some task $a$ is to predict whether a visual instance 'has wheels' or not, and task $b$ is to predict if a given visual object 'is fast'; not only is one easier, but there is also a dependence between them. So there is a possibility that easier tasks reach their best validation metric before the rest and may start overfitting if training were to be continued. This isn't reflected very clearly with the use of a validation metric that is averaged over all tasks. As a larger number of underfit tasks would skew the average, the overall optimal validation point gets shifted to a later time-step (epoch) when the model could be worse at the easier tasks. Vice versa, the optimal epoch gets shifted earlier due to a larger, easier subset that are overfit when the harder tasks reach their individual optimal epochs. We term this mismatch in the overall and task optimal epochs as a 'temporal discrepancy'.

In this work, we explore and try to mitigate this discrepancy between tasks. We present in this paper that early stopping on only the expected error over tasks leaves us blind to the performance they are sacrificing per task. The work is organized in the following manner: in §2, we explore existing work that deals with methods for incorporating task difficulty (which could be causing this discrepancy) into training. The rest of the sections along with our contributions can be summarized as:

1. We present a method to easily visualize and detect the discrepancy through *interval plots* in §3

2. We formulate techniques that could quantify this discrepancy by also considering the per-task validation metrics in model selection in §4.

3. We explore the presence of the temporal discrepancy on three image datasets and test the aforementioned techniques to assess the change in performance in §5

4. To the best of our knowledge, there has not been a study like this into the potential of per-task validation metrics to select an ensemble of models.

## 2 RELATED WORK

Training multiple related tasks together creates a shared representation that can generalize better on individual tasks. The rising prominence of multi-task learning can be attributed to Caruana (1997). It has been acknowledged that some tasks are easier to learn than the others and plenty of works have tried to solve this issue through approaches that slow down the training of easier tasks. In other words, tasks are assigned a priority in the learning phase based on their difficulty determined through some metric. This assignment of priority implicitly tries to solve the temporal discrepancy without formally addressing its presence. Task prioritization can take the form of gradient magnitudes, parameter count, or update frequencies (Guo et al., 2018). We can group existing solutions into task prioritization as a hyperparameter or task prioritization during training (aka self-paced learning). The post-training brute force and clustering methods we propose do not fit into these categories as we believe they have not been done before. Instead of adjusting training or retraining, these methods operate on a model which has already been trained.

**Task prioritization as a hyperparameter** is a way to handle per task overfitting that is almost the subconscious approach for most practitioners. This would include data-augmentation and over/undersampling. An example case is in Kokkinos (2017) where they use manually tuned task weights in order to improve performance.

**Task prioritization during training** covers approaches where tasks dynamically change priority or are regularized in some way. For example Guo et al. (2018) takes an approach to change task weights during training based on multiple metrics such as error, perceived difficulty, and learnable parameters. The idea is that some tasks need to have a high weight at the start and a low weight later in training. In a similar direction Gradnorm (Chen et al., 2018) aims to set balance task weights based on normalizing the gradients across tasks.

**Using relationships between tasks** during training is another direction. Ruder (2017) discussed *negative transfer* where sharing information with unrelated tasks might actually hurt performance. Work by Lee et al. (2016) incorporated a directed graph of relationships between tasks in order to enforce sharing between related tasks when reweighting tasks. Task clustering has been performed outside of neural networks by Evgeniou et al. (2005); Evgeniou & Pontil (2004) where they regularize per-task SVMs so that the parameters between related tasks are similar.

It would be natural to use some of these methods as a baseline for our work. However, we think it would not be an equitable comparison as:

- These baseline methods are applied during training whereas ours is a post-training analysis.

- The main aspect of our analysis is only on the validation metric whereas these baselines consider a variety of different aspects of training.

- The focus of our work is on how the weights change with time, keeping all else constant, and how these changes affect the validation and test performance. The aforementioned methods modify the gradients w.r.t. several factors during the training which adds more degrees of freedom and is difficult to compare.

Regardless of task difficulty, training multiple tasks jointly with a neural network can lead to *catastrophic forgetting*: it refers to how a network can lose information that it had learned for a particular task as it learns another task (McCloskey & Cohen, 1989). Multiple works have explored and tried to mitigate this phenomenon (Ratcliff, 1990; Robins, 1995; Goodrich & Arel, 2014; Kirkpatrick

et al., 2017; Kemker et al., 2018; Lee et al., 2017) and it still remains an open area of research. It is highly likely that catastrophic forgetting could be causing any such temporal discrepancy; exploring the relationship between the two is an area is a very interesting direction in research and is left for future work.

# 3 STUDYING TEMPORAL DISCREPANCY BETWEEN TASKS

Firstly, we define what a task is to disambiguate from its general usage in multi-task learning literature. A 'task' is predicting a single output unit out of many, regardless of the training paradigm being multi-class or multi-label or other. Tasks can be very fine-grained such as predicting the class of an image or much higher-level such image classification, segmentation etc. While our work uses the term in the former context, our motivation and findings can be applied in the latter context (which is the broader and more common context in multi-task learning) as well.

In the next two subsections, we define the term temporal discrepancy and display an example of it on CIFAR100. Then, we introduce a simple method of visualizing it on datasets with a large number of tasks that would make it difficult to analyze the per-task curves together.

## 3.1 TEMPORAL DISCREPANCY

A temporal discrepancy in the validation performance refers to the phenomenon where the model isn't optimal for all of its tasks simultaneously. This occurs when there is a difference between the overall optimal epoch determined by the summarized validation metric and the epoch in which task achieves its best validation metric is higher than some arbitrary threshold, i.e., $|t_s - t_i| > \delta$ where $t_s$ is the optimal epoch of the summarized validation curve and $t_i$ is the optimal epoch for task $i$.

Figure 1 displays an example of this discrepancy in CIFAR100 (only five curves plotted for clarity). It is most evident for the labels *Sea* and *Lamp* which undergo a drop of 7.5% and 5.7% respectively in their validation accuracy from their peak epoch to $t_s$. Similarly, *Snake* also starts degrading till $t_s$ but strangely starts improving after. Conversely, *Rose* and *Streetcar* are underfit at $t_s$ as they continue to improve after.

The most noteworthy observation is that the averaged validation curve (in dotted black) completely plateaus out after the $150^{th}$ epoch. There is significant variation occurring in the per-label curves but the averaged curve is unable to represent these dynamics in the training. Selecting an optimal model off the averaged curve can be quite misleading as it represents the entire $[151, 300]$ interval as optimal despite the labels' validation accuracies fluctuating significantly in this interval. The test performance of individual labels can wildly differ depending on which epoch is used for loading the weights for testing and/or deployment.

## 3.2 INTERVAL PLOTS

It is easy to examine the per-label curves in Figure 1 as only 5% of the labels have been plotted. But when the number of tasks is high and all of them need to be plotted together to get a clearer global picture, decomposing the summarized validation curve can get very messy. *Quasi-optimal validation interval plots*, or interval plots for short, are a way of assessing the optimal temporality of the per-task validation performance relative to $t_s$. It is a simple visualization method that aids in determining when and/or for how long the tasks are within the acceptable limits of the best validation performance and also which and/or how many tasks aren't within these limits near the overall optimal epoch $t_s$.

Creating an interval plot involves finding a 'quasi-optimal' region for each task, i.e., a consecutive temporal interval in which a validation metric of the task fluctuates near its maximum with a set tolerance. The task validation curves are first smoothed out to reduce noise and the time-step (epoch) at which the task achieved its optimal validation metric is determined. Then, the number of epochs before and after this task-optimal epoch in which the task metric is greater than a threshold is calculated. This duration of epochs is **the interval** for the task.

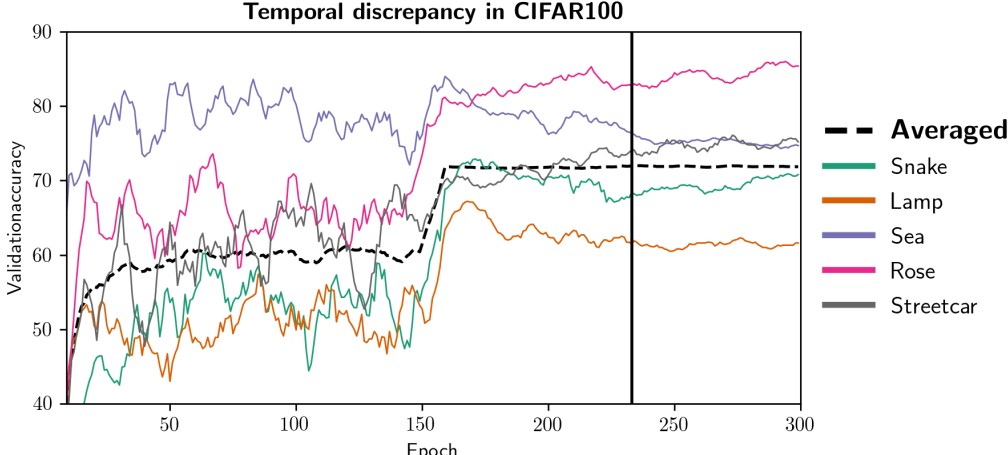

Figure 1: Visualizing the presence of a temporal discrepancy in five of CIFAR100 labels (training is detailed in Section 5). The vertical black line is the best overall epoch $t_s$. The dotted and solid curves represent the averaged validation curve and the label specific validation curves respectively. All curves have been smoothed by averaging over a sliding window of size 10.

Given a vector of validation metrics $\mathbf{A_i}$ for a task $i$, its interval $\boldsymbol{\tau_i}$ is given by:

$$\boldsymbol{\tau_i} = [t_i - m, \ldots t_i - 1, t_i, \ldots t_i + n] \quad \forall \, a_{ij} \geq a_{it_i} - \epsilon$$
$$\text{where } t_i = \text{argmax } \mathbf{A_i}, \; j \in \boldsymbol{\tau_i} \text{ and } a_{ij} \in \mathbf{A_i}$$

Figure 2 plots the decomposed curves and the equivalent intervals for CIFAR100. The overall optimal epoch $t_s$ doesn't fall in the intervals of almost half the labels; these labels aren't at their potentially best validation performance at the early stopping point. Some intervals are notably small in duration, meaning those labels have a very sharp peak. This could imply that the validation performance is randomly high at that epoch and it'd be more suitable to shift the quasi-optimal region of these labels to a longer and/or later interval, that doesn't necessarily contain $t_i$, as long as the validation accuracy stays within the tolerance in that interval.

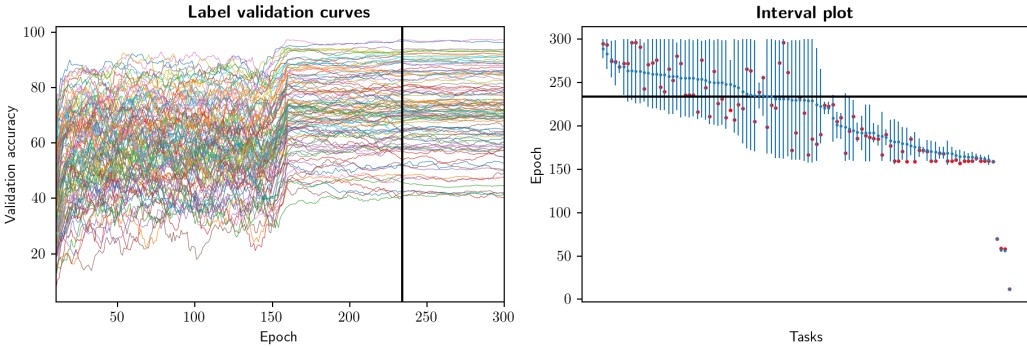

Figure 2: Smoothed decomposed validation curves (left) and the equivalent interval plot (right) for CIFAR100 with $\epsilon = 2\%$. For both plots, the straight black line is the best overall epoch $t_s$. On the interval plot, the blue lines indicate the interval $\boldsymbol{\tau_i}$, the blue dots are the centers of the interval, and the red dots are the task-optimal epoch $t_i$. The intervals have been sorted by their centers

## 4 QUANTIFYING PERFORMANCE LOST DUE TO TEMPORAL DISCREPANCY

In this section, we present two simple techniques that consider the per-task validation metrics for selecting the best model for testing. Two aspects common to these techniques is their hand-crafted

& engineered nature and their inefficiency in terms of deployment, training time and/or inference time. The aim with these techniques to assess how much potential gain in performance could be attained if we account for per-task validation metrics in selecting a model and we'd like to stress that these techniques serve as a proof-of-concept of this gain (if any). They are intended as a baseline and also a stimulus for increasing research into the effect of the subtleties of the validation curves on model performance and selection.

**Brute force**   This involves loading the model with the weights from a given task $i$'s optimal validation epoch $t_i$ and evaluating on only the samples that belong to that task. We call this particular model that has been loaded with the weights from $t_i$ as the validation-optimal model for task $i$. It is essentially using a "separate model" for each task during evaluation and/or deployment making this the most naive approach. It is also the most inefficient because it would (i) require storing up to $N$ models, where $N$ is the total number of tasks (ii) require a way to combine predictions from all $N$ models that wouldn't be misleading during inference (iii) increase latency significantly due to overhead caused by loading and reloading the model weights (iv) scale up the inference time by a factor up to $N$.

**Clustering**   Instead of having separate weights for each task, we try to cluster the set of the task-optimal validation epochs into $K$ clusters so that only $K$ models are required as opposed to $N$. In this approach, the interval plots can also be utilized to cluster tasks that have similar interval positions and/or lengths in addition to the $t_i$'s. Similar to brute force, this technique also involves multiple models loaded with weights from different epochs, but trades off any gain in performance for a lower number of models.

## 5 EXPERIMENTS

We train variations of DenseNets (Huang et al., 2017) on three image datasets: CIFAR100, Tiny ImageNet and PadChest. All models were trained with three random seeds for model initialization and splits of the training set into training and validation sets.

**CIFAR100**   CIFAR100 (Krizhevsky & Hinton, 2009) is a dataset of natural images containing 100 classes with 500 training images and 100 testing images per class. We trained a "DenseNet-BC (k = 12)" as described in Huang et al. (2017): it has three dense blocks, a total of 100 layers and a growth rate of 12. It was trained in the exact manner as the original work, i.e. 300 epochs with dropout, weight decay of $10^{-4}$, SGD with Nesterov momentum of 0.9 and an initial learning rate of 0.1 that is divided by 10 at the $150^{th}$ and $225^{th}$ epochs. As we had carved out 20% of the training set as validation and didn't use data augmentation, we achieved an average test accuracy of 72.14%. In our analysis, we only use the validation curves after the $150^{th}$ epoch because the training is very noisy and brittle up to that epoch due to the use of a learning rate of 0.1 (Figures 1 and 2).

**Tiny ImageNet**   ImageNet (Deng et al., 2009) is a dataset with 1.5 million images and 1000 classes. Tiny ImageNet[1] is a subset of ImageNet with images resized to 64x64 and only 200 classes. We utilized the same architecture as CIFAR100 but with a total of 190 layers and a growth rate of 40. In addition, we used a stride of 2 for the first convolution. This modified DenseNet was also trained in the same manner as above but with the hyperparameters used for ImageNet in the original work: 100 epochs, no dropout and dividing the learning rate at the $30^{th}$ and $60^{th}$ epochs instead. Rest of the hyperparameters remained the same. 10% of the training set was used as validation and we achieved an average test accuracy of 63.09%. Similar to CIFAR100, we only use the validation curves after the $30^{th}$ epoch. The interval plot from one run of training is given in Figure 3a

**PadChest**   PadChest (Bustos et al., 2019) is a medical imaging dataset of 160,000 chest X-rays of 67,000 patients with multiple visits and views available. We used the publicly available code provided by Bertrand et al. (2019) to recreate their cohort of around 31,000 samples. We trained a multilabel DenseNet-121 (Huang et al., 2017; Rajpurkar et al., 2017) on the frontal views and only those labels which have more than 50 samples (total 64 labels). With a 60-20-20 split between training, validation, and test sets, we trained for 100 epochs with Adam and an initial learning rate of

---

[1]https://tiny-imagenet.herokuapp.com/

0.0001 that is halved every 20 epochs. The interval plot from one run of training is given in Figure 3b

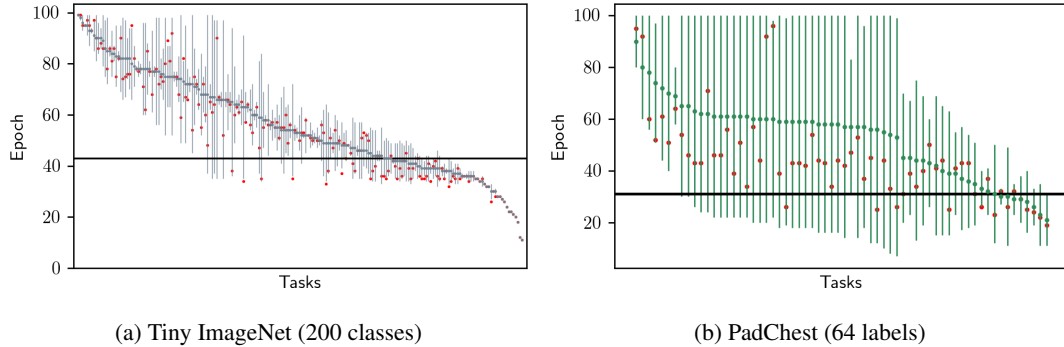

(a) Tiny ImageNet (200 classes)                    (b) PadChest (64 labels)

Figure 3: Interval plots for Tiny ImageNet and PadChest with $\epsilon = 0.02$. The gray/green lines and dots indicate the interval $\tau_i$ and its center respectively. The black horizontal line is the best overall epoch $t_s$ and the red dots are the task-optimal epoch $t_i$. The intervals have been sorted by their centers

## 5.1 BRUTE FORCE

By brute forcing the best model selection, we wanted to assess how much performance is lost when the summarized validation curve is used. This involves evaluating each task with its own set of optimal weights determined from its specific optimal epoch. This would require *N* models in theory but the number of validation-optimal models is actually lower as many tasks have inter-dependent learning profiles. These correlated tasks may reach their optimal validation performance at the same epoch, thus requiring only a single common set of weights for all of them. *N* also decreases with the total number of training epochs as that increases the probability of tasks having the same optimal epoch. On analyzing the task validation curves for the three datasets, we do find that the number of models required is much lower than *N*. CIFAR100 and Tiny ImageNet both required less than 60 models for brute forcing the evaluation, despite the latter having twice as many labels. Also since Tiny ImageNet was trained for one-third of the epochs, the number of models reduced drastically. The number of models for PadChest was around 35.

The results of using a single model for each task are tabulated for the three datasets in Table 1. On using each label's validation-optimal model for evaluating on the test, the test metric always increases in comparison to using the baseline model that uses the summarized validation curve for CIFAR100 and Tiny ImageNet. The top-1 accuracy for CIFAR100 undergoes an average and maximum increase of approximately 2.5% & 3.2% respectively. For Tiny ImageNet, an average and maximum increase of 1.72% & 1.95% is observed.

However, the average increase in PadChest is not only meagre but also within the standard deviation. This could be because the temporal discrepancy isn't very pronounced in PadChest: the overall

Table 1: Accuracy/AUC on the test set for the baseline and brute forced models, averaged over 3 seeds

| Dataset | Metric | Baseline | Brute-force |
|---|---|---|---|
| CIFAR100 | Top-1 | $72.14 \pm 0.51$ | **$74.62 \pm 0.43$** |
|  | Top-5 | $92.23 \pm 0.33$ | **$93.07 \pm 0.26$** |
| Tiny ImageNet | Top-1 | $63.09 \pm 0.27$ | **$64.82 \pm 0.39$** |
|  | Top-5 | $83.79 \pm 0.26$ | **$84.91 \pm 0.47$** |
| PadChest | AUC | $0.7807 \pm 0.001$ | $0.7826 \pm 0.002$ |

optimal epoch does happen to be in the intervals of a very large portion of the labels (Figure 3b). As PadChest has fewer number of outputs compared to the other two (64 vs. 100/200), there could also be a relationship between the number of outputs and the magnitude of the discrepancy's effect. These results show that accounting for the per-task validation metrics can mitigate a significant temporal discrepancy and increase the testing performance.

## 5.2 COMPUTATION-PERFORMANCE TRADEOFF

We can further reduce the number of validation-optimal models required by merging the $t_i$'s that are close to each other. We use K-means to cluster the set of task-optimal epochs into $K$ clusters. We vary $K$ from 2 to $N$ to gain an insight into how much performance is lost as we reduce the number of validation-optimal models required down to one where using only one model is the same as using the summarized. For any given cluster $k$, we use its center as the epoch for loading the weights to evaluate the set of the tasks in $k$. We round off the center to the nearest integer for our analysis but the fractional part can be used to select weights after a specific batch iteration in that epoch.

The results for the three datasets are plotted in Figure 4. There is an interesting observation that the accuracy doesn't always increase with the number of task-optimal models which is very noticeable in Tiny ImageNet Seed 2. This could be due to the noisy nature of training with SGD where the validation metric can oscillate a lot between epochs and even a shift of one epoch can cause a decrease in performance.

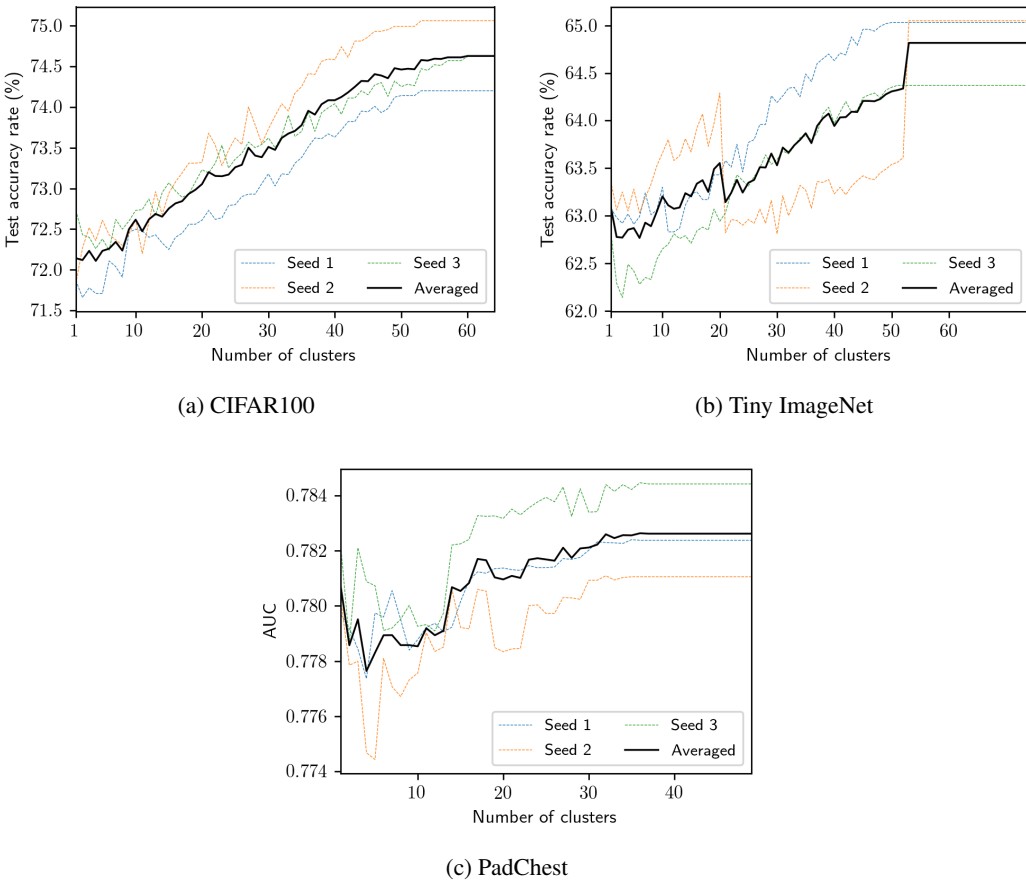

(a) CIFAR100

(b) Tiny ImageNet

(c) PadChest

Figure 4: Reducing the number of models needed to represent all per-task optimal validation points. $K$ is the number of clusters

## 6 CONCLUSION

In this work, we examine the decomposition of a model's average validation curve into its per-task curves to assess the presence of a temporal discrepancy on three image datasets. We provide a visualization method to detect if there is a disparity between the task curves and the summarized validation curve. We test two techniques that incorporate the per-task metrics into model evaluation and we find that that when per-task validation metrics are accounted for training runs that show a significant temporal discrepancy, we gain an increase in testing performance. We show that with CI-FAR100 there is room for up to 2.5% to be gained in testing accuracy and 1.72% for Tiny ImageNet if we don't use the averaged metrics as they are.

With these experiments, we aim to create more awareness of how summarized validation metrics cannot represent a model that is truly optimal for all its tasks all the time. Using averaged curves could mean that models are currently being trained oblivious of the performance being sacrificed on individual tasks. We wish to draw attention to the need of both theoretical and engineered approaches that would take the per-task validation metrics into account while training. Our experiments demonstrate that there is a potential that current state-of-the-art models could possibly be made even more optimal by ensuring all of its prediction tasks are at their validation optimums.

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
