# OpenReview forum: "On summarized validation curves and generalization"
_ICLR.cc/2020/Conference — Reject_

### Official Review · AnonReviewer3 · 2019-10-17
**Official Blind Review #3**

**Rating:** 3

**Review:**

The paper examines the common practice of performing model selection by choosing the model that maximizes validation accuracy. In a setting where there are multiple tasks, the average validation error hides performance on individual tasks, which may be relevant. The paper casts multi-class image classification as a multi-task problem, where identifying each different class is a different task.

At different points of training, a model's performance on a task will vary. To make it easier to examine this, they propose displaying an interval plot. The validation accuracy for a single task will be largest at a specific epoch. For each single task, display the epochs where performance is within some threshold of the best validation-set accuracy for that task. Plots on CIFAR-100 demonstrate how some tasks are learned quickly then forgotten, while other tasks are only learned late.

To examine the difference in performance, they compare the single best CIFAR-100 model, to the classifier defined by "for class k, forward input to the model with best validation accuracy for class k". This gives about +2.5% in Top-1 performance, a similar story appears on the other 2 datasets they try (Tiny ImageNet, PadChest). This requires N models, where N is the number of classes. They also examine a clustering approach, where they cluster the N models into K models (using K-means), then forward class k to the best model out of those K models.

Overall, it isn't an especially large contribution but I felt it had some interesting points. Some comments:
* Citation list feels short. The catastrophic forgetting literature seems relevant here. Rich Caruana's multitask learning thesis (Caruana et al 1997) is also relevant.
* It's unclear how the K-means is carried out. I assume the features used for each model checkpoint are tied to its performance on individual task but it's never spell out what distance metric is used in the K-means.
* It feels strange to have no comparisons to ensemble-based baselines. The baseline here would be, for some parameter K, run K independent training runs, take the top average validation accuracy from each run, and average them together, then compare this to the K models found from K-means clustering. For small enough K (like 5 or 10) this seems like a feasible experiment, computation-wise.
* In a similar vein, I wonder how this compares to the Snapshot Ensembles paper, which has a similar guarantee of doing 1 training run and giving M models for an ensemble. Is the gain from ensembling, or from directly examining which models are better at which classes?
* A natural follow-up here is to use model distillation to distill the N best models for each individual class into a single model checkpoint - does this give us a better single model checkpoint?

As-is, I give this paper a weak reject, but would be willing to increase if some of these experiments were tried.

Edit: having read the other reviews and author comments, I still maintain a weak reject rating. I believe this paper needs further work.

**Experience Assessment:**

I have read many papers in this area.

**Review Assessment: Checking Correctness Of Derivations And Theory:**

I carefully checked the derivations and theory.

**Review Assessment: Checking Correctness Of Experiments:**

I assessed the sensibility of the experiments.

**Review Assessment: Thoroughness In Paper Reading:**

I read the paper thoroughly.

---

> ### Author Response · Authors · 2019-11-13
> **Thank you for your comments**
>
> > Citation list feels short. The catastrophic forgetting literature seems relevant here. Rich Caruana's multitask learning thesis (Caruana et al 1997) is also relevant.
>
> We’ve added the relevant literature.
>
> > It's unclear how the K-means is carried out. I assume the features used for each model checkpoint are tied to its performance on individual task but it's never spell out what distance metric is used in the K-means.
>
> We use each task’s optimal valid epoch t_i as the distance metric
>
> > It feels strange to have no comparisons to ensemble-based baselines. The baseline here would be, for some parameter K, run K independent training runs, take the top average validation accuracy from each run, and average them together, then compare this to the K models found from K-means clustering. For small enough K (like 5 or 10) this seems like a feasible experiment, computation-wise.
>
> This does happen to be our baseline, with K=3. The numbers in Table 1 have been averaged over 3 independent runs of different train/valid splits and model initializations. However, the test set for CIFAR100 and TinyImagenet remained the same for all runs. On examining Fig 4, we expect it to be worse than the baseline possibly due to the noisy nature of SGD training (which we address in Sec 5.2, second paragraph).
>
> > In a similar vein, I wonder how this compares to the Snapshot Ensembles paper, which has a similar guarantee of doing 1 training run and giving M models for an ensemble. Is the gain from ensembling, or from directly examining which models are better at which classes? A natural follow-up here is to use model distillation to distill the N best models for each individual class into a single model checkpoint - does this give us a better single model checkpoint?
>
> We are hesitant to include baselines that involves modifying the training procedure, as our analysis is done post-training so we don’t think it’d be an equitable comparison. We’ve elaborated on our reasons in more detail in response to Reviewer#2

---

> > ### Comment · AnonReviewer3 · 2019-11-13
> > **Thanks for your reply**
> >
> > Thanks for your reply. Although the comparisons aren't directly equitable, I don't think this means you should avoid them entirely, as inequitable comparisons still provide signal on why your proposed approach helps.
> >
> > I believe the Table 1 numbers compare K=3 seeds of regular training and K=3 seeds of brute-force comparison. What I wanted was 3 seeds of regular training, defining the ensemble by the average of their predictions, then comparing this to a K-means reduction of the models, rather than to the brute force model. This ensemble averaging is different from just averaging the accuracy of the 3 random seeds (we expect the ensemble of 3 to outperform any individual model.)

---

> > > ### Author Response · Authors · 2019-11-15
> > > **Thank you**
> > >
> > > Thank you for your reply. From your comments, we do think now that the paper would benefit from including these baselines but it won't be possible to update this paper due to the revision deadline. Do you have any more suggestions on baselines, especially any that focus particularly on the validation accuracy?
> > >
> > > In your definition of an ensemble, can you clarify this small thing: is it that the train/val split remains the same but the model initialization is different between the training runs? Or it doesn't matter?

---

### Official Review · AnonReviewer2 · 2019-10-22
**Official Blind Review #2**

**Rating:** 3

**Review:**

Summary
Model validation curve typically aggregates accuracies of all labels. This paper investigates the fine-grained per-label model validation curve. It shows that the optimal epoch varies by label. The paper proposes a visualization method to detect if there is a disparity between the per-label curves and the summarized validation curve. It also proposes two methods to exploit per-label metrics into model evaluation and selection. The experiments use three datasets: CIFAR 100, Tiny ImageNet, PadChest.

Limitations
The paper is very preliminary in nature. It does not compare with other related work. For example,  the task prioritization during training approach where tasks dynamically change priority or are regularized in some way. It will be good to see how the proposed approach compare with other related ones (three listed in related work) in the literature.

Other approaches not mentioned in the paper can be more effective. For example,
Focal Loss for Dense Object Detection, https://arxiv.org/abs/1708.02002
might automatically handle the problem to a large extent.

How much the problem is due to label imbalancing? If label imbalancing is one main problem, please first address it.

The proposed approach is not very interesting, brutal force and clustering. They are very straightforward.

Overall, the quality is far below the bar of ICLR.

Comments not affect rating

The paper uses "task" which is not defined. It is confusing until much later in the paper that it just refers to "class".


**Experience Assessment:**

I have read many papers in this area.

**Review Assessment: Checking Correctness Of Derivations And Theory:**

N/A

**Review Assessment: Checking Correctness Of Experiments:**

I carefully checked the experiments.

**Review Assessment: Thoroughness In Paper Reading:**

N/A

---

> ### Author Response · Authors · 2019-11-13
> **Thank you for your comments**
>
> We tried to include more baselines and ultimately chose not to because:
> (i) These baseline methods are done during training and (most) don’t involve validation metrics as the main aspect of their approaches. We perform our analysis post-training with all the focus on validation metrics.
> (ii) We only wanted to consider how the weights change with time keeping all else constant and how these changes affect the validation and test performance. As these baseline methods modify the gradients wrt several factors during the training, it would add more degrees of freedom and would be difficult to compare.
> Because of these two reasons, it felt it wouldn’t be an equitable comparison. Moreover, we’ve been unable to find any research papers in this niche that we could implement as a baseline.
>
> > How much the problem is due to label imbalancing?
> There is no imbalance in CIFAR100 and TinyImagenet. PadChest does have an imbalance, but the network had been trained with class weights.
>
> > The proposed approach is not very interesting, brutal force and clustering. They are very straightforward.
> Yes, we do remark on the naive and inefficient nature of the two methods we use. We’d like to state that these methods themselves aren’t intended as a contribution; rather their results are used to develop our main contribution: that simply using averaged metrics doesn’t guarantee the best testing performance and there is a potential increase if per-task validation metrics are used. We’ve revised the paper to make sure this is reflected better.
>
> Comments not affect rating
> The paper uses "task" which is not defined.
> We mention our definition of a task in the footnote of Page 2. We’ve moved it to the beginning of the section
>
> We believe the main reason for your rejection decision was due to the work not including necessary baselines. As we explained above, we believe there are no equitable comparisons. Please let us know if you do not agree.

---

### Official Review · AnonReviewer1 · 2019-10-26
**Official Blind Review #1**

**Rating:** 1

**Review:**

Summarize what the paper claims to do/contribute. Be positive and generous.
The paper proposes a new method for model selection in the case of classification with multiple labels. The method suggests not relying on average accuracy over all labels to choose a model but choosing multiple models depending on the label and apply them to the related samples.

Clearly state your decision (accept or reject) with one or two key reasons for this choice.
Reject.
- I don't think the paper is contributing something new to the literature. Also there seems to be some confusion re: what accuracy means and is used in some places instead of precision.  (eg in. Sect. 4.1)

Provide supporting arguments for the reasons for the decision.
- One question I had was to think about is how one would know what model to apply to what samples if they don't know a priori the labels of these samples. This seems to be something your suggested approaches rely on.

Provide additional feedback with the aim to improve the paper. Make it clear that these points are here to help, and not necessarily part of your decision assessment.
- "valid" is used often instead of validation.


**Experience Assessment:**

I have published in this field for several years.

**Review Assessment: Checking Correctness Of Derivations And Theory:**

I assessed the sensibility of the derivations and theory.

**Review Assessment: Checking Correctness Of Experiments:**

I carefully checked the experiments.

**Review Assessment: Thoroughness In Paper Reading:**

I read the paper thoroughly.

---

> ### Author Response · Authors · 2019-11-13
> **Thank you for your comments**
>
> We believe that the contribution of the paper has been misunderstood. The main message of the paper is that using a summarized validation metric for a given purpose tends to leave out nuances in how each task has been performing throughout training. We demonstrate that accounting for these nuances can lead to better model selection and hence a better test performance through two simple methods. We’d like to stress two things -
>
> (i) The two methods themselves are not a contribution; rather we use the results of these methods to support the contribution
>
> (ii) Model selection is the specific use case we chose for this paper to test our hypothesis. Our contribution can also apply/be tested for other purposes, such as reducing the learning rate based on the validation metric.
>
> > Also there seems to be some confusion re: what accuracy means and is used in some places instead of precision.  (eg in. Sect. 4.1)
>
> This is unclear, where should we have used the word precision?
>
> > One question I had was to think about is how one would know what model to apply to what samples if they don't know a priori the labels of these samples. This seems to be something your suggested approaches rely on.
>
> This is a very valid question and a major shortcoming of the two methods. To answer the question, it depends on what level of abstraction of task are the multiple models based on. If prediction tasks are independent, such as in the case of multi-label (like PadChest), then every model needs to be applied. It is similar for the general definition of the multi-task setting (eg. classification, segmentation etc). But for multi-class, it’d be difficult to determine. One could extract just the logit of the class produced by its particular valid-optimal model and take the softmax over the vector of these “valid-optimal” logits.
> However, we’d like to reiterate that our suggested approaches serve as only a proof-of-concept of the gain in performance when per-task metrics are accounted for. These methods aren’t intended to be adopted in current practice but their results are intended as a “stimulus for increasing research” in the area of examining the subtleties in validation curves w.r.t. individual tasks and also as a “baseline” for any future work in this area.
>
> We understand there was ambiguity in the paper regarding the exact contribution. We hope our response makes it more clear and we have revised the paper to make sure it isn’t ambiguous anymore. Have we addressed all of your concerns?

---

### Decision · Program_Chairs · 2019-12-19

**Decision:**

Reject

**Comment:**

The reviewers reached a consensus that the paper is preliminary and has a very limited contribution. Therefore, I cannot recommend acceptance at this time.